# Periodic Bayesian Flow Networks with Additive Accuracy

**Peijia Lin** [* 1 2]   **Zihan Zhang** [* 3 2]   **Zhangrui Zhao** [4]   **Shaohao Rui** [5 2]   **Junyi An** [6]   **Yunfei Shi** [6]   **Fenglei Cao** [6]
**Weijie Ma** [7 2]   **Yutong Lu** [1 8]

## Abstract

Generating periodic data—such as fractional atomic coordinates in crystal structures and phase patterns in compressive light-field (CLF) displays—is challenging because wrap-around boundaries complicate probabilistic modeling and learning. While Bayesian Flow Networks (BFNs) offer a powerful generative framework with strictly additive accuracy in Euclidean space, existing periodic adaptations typically sacrifice additivity and become sensitive to schedule heuristics. We introduce *PeriodicBFN*, which embeds each periodic scalar into a two-dimensional unit-circle representation and performs Gaussian Bayesian updates in the resulting Cartesian space, thereby restoring strictly additive accuracy. To address invariance in periodic generative modeling, we further derive a Rao–Blackwellized objective that analytically marginalizes global periodic translations, producing a translation-invariant target with reduced gradient variance. Experiments on crystal structure prediction and multi-layer phase synthesis for CLF displays demonstrate improved training stability and strong performance. To our knowledge, this is the first work to extend periodic-data generative modeling to phase synthesis for modern glasses-free 3D display systems.

## 1. Introduction

Deep generative models (Ho et al., 2020; Song et al., 2021; Lipman et al., 2023; Graves et al., 2023) have become central tools in AI for Science, with growing impact in molecular design (Song et al., 2024; Tao & Abe, 2025), protein generation (Atkinson et al., 2025), materials science (Xie et al., 2022; Jiao et al., 2023), and computational displays (Ma et al., 2025). In materials science and optics, data-driven generative methods increasingly complement compute-intensive physics-based pipelines: crystal structure prediction (CSP) has progressed from first-principles and heuristic search (Glass et al., 2006; Wang et al., 2010) to diffusion, flow-matching, and BFN-based generation (Xie et al., 2022; Jiao et al., 2023; Lin et al., 2024; Miller et al., 2024; Wu et al., 2025), while compressive light-field (CLF) displays such as EyeReal (Ma et al., 2025) use learned phase synthesis for real-time glasses-free 3D display, a visual interface for interactive 3D content (Kara & Simon, 2023), spatial computing, and digital twins (Wang et al., 2025).

A common mathematical thread underlying these two applications is *periodic data* under wrap-around boundaries. Crystal structures are naturally represented by fractional atomic coordinates modulo one (Jiao et al., 2023), while CLF devices require generating multi-layer *phase patterns* whose values are defined modulo $2\pi$ (Ma et al., 2025). These periodic manifolds (e.g., tori) break naive Euclidean parameterizations, creating discontinuities that complicate probabilistic modeling and learning.

Most existing approaches address periodic data through dynamical generative processes that gradually transport samples toward the data manifold, such as diffusion models (Jiao et al., 2023) and flow matching (Miller et al., 2024). Recently, Bayesian Flow Networks (BFNs) (Graves et al., 2023) have emerged as an alternative paradigm that evolves *distributions* via repeated Bayesian conditioning. In crystal generation, BFN adaptations (e.g., CrysBFN (Wu et al., 2025)) model periodic coordinates using low-entropy circular distributions (e.g. von Mises) and iteratively assimilate network-predicted observations in parameter space, often achieving strong sample quality and sampling efficiency.

However, current circular adaptations remain misaligned with a core BFN design principle: *strictly additive accuracy*

---

[*]Equal contribution  [1]School of Computer Science and Engineering, Sun Yat-sen University [2]Shanghai Innovation Institute [3]College of Excellence Engineers, Nankai University [4]School of Computer Science and Engineering, Beihang University [5]School of Information and Electronic Engineering, Shanghai Jiao Tong University [6]Shanghai Academy of Artificial Intelligence for Science [7]College of Computer Science and Artificial Intelligence, Fudan University [8]National Supercomputer Center in Guangzhou. Correspondence to: Weijie Ma <weijiema@sii.edu.cn>, Yutong Lu <luyutong@mail.sysu.edu.cn>.

*Proceedings of the $43^{rd}$ International Conference on Machine Learning*, Seoul, South Korea. PMLR 306, 2026. Copyright 2026 by the author(s).

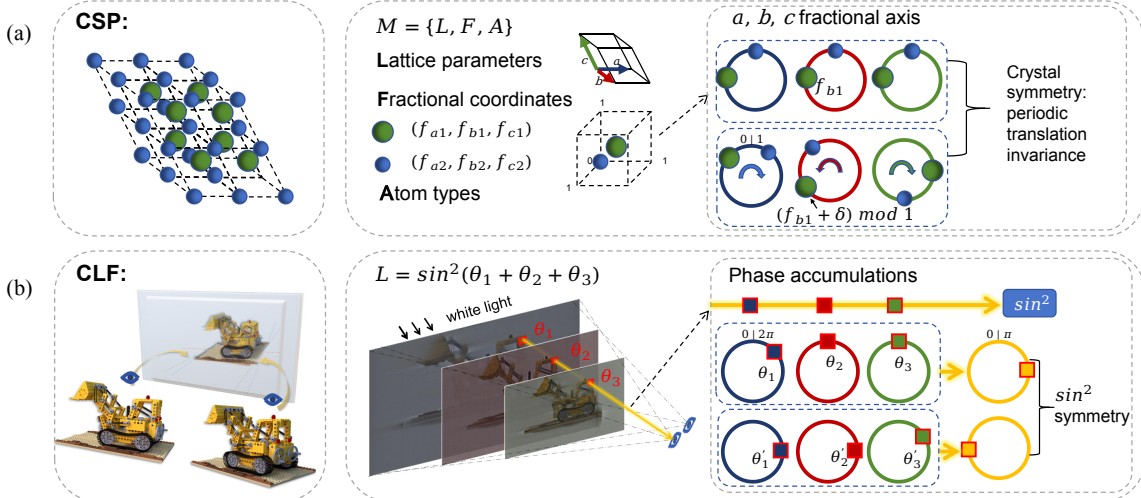

*Figure 1.* Application scenarios and periodic-data structures. **(a) Crystal Structure Prediction (CSP)**: Fractional coordinates $F \in [0, 1)^{N \times 3}$ inherently reside on circular manifolds, exhibiting continuous global periodic translation invariance under uniform shifts. **(b) Compressive Light-Field (CLF) Phase Synthesis**: Multi-layer phase accumulations are mapped to emitted intensities via Malus's law ($L = \sin^2(\sum \theta_n)$). This induces an intrinsic $\pi$-periodic symmetry (sign equivalence), where distinct phase states map to identical observations.

across the iterative Bayesian updates. For circular families such as von Mises, concentration updates are not generally additive, so the effect of each assimilation step can depend strongly on the quality of the predicted observation. This sensitivity motivates additional heuristics and scheduling strategies (Wu et al., 2025) and can lead to unstable generation.

Moreover, periodic data often exhibit *periodic translation invariance*. For crystals, translating all fractional coordinates by a global periodic translation yields an equivalent structure (Jiao et al., 2023); for CLF phase synthesis, global periodic translations can likewise induce nuisance symmetries depending on the forward model. Enforcing invariance only at inference time while training against non-invariant targets creates an objective mismatch that inflates gradient variance (Lin et al., 2024).

To address these issues, we develop *PeriodicBFN*, a periodic-data generative framework that maps each periodic scalar onto the unit circle in $\mathbb{R}^2$ and performs Gaussian Bayesian updates in Cartesian space. This preserves the original BFN additive-accuracy structure, yields a clean continuous-time loss, and lets the neural network learn geometric coupling on the embedded circle.

We further derive a Rao–Blackwellized objective that analytically marginalizes periodic nuisance symmetries, producing invariant targets with reduced gradient variance. Together, these components yield an E(3)-equivariant periodic BFN for crystal generation and extend periodic-data generative modeling to CLF phase synthesis.

Our contributions are summarized as follows:

- **Additive-accuracy periodic BFN:** we introduce a 2D unit-circle embedding that permits Gaussian Bayesian updates for periodic variables, restoring the strictly additive accuracy property of Euclidean BFNs.

- **Invariant training by Rao–Blackwellization:** we derive closed-form Rao–Blackwellized objectives that marginalize nuisance periodic symmetries, yielding translation- or symmetry-invariant targets with reduced gradient variance.

- **Cross-domain validation:** we instantiate the framework for E(3)-equivariant crystal structure prediction and CLF phase synthesis, demonstrating stable generation and strong empirical performance in both materials science and glasses-free 3D display applications.

## 2. Related Work

**Crystal structure generation.** Crystal structure prediction has progressed from physics-based and heuristic search procedures to neural generative modeling with symmetry- and equivariance-aware backbones (Xie et al., 2022; Jiao et al., 2023; Lin et al., 2024). Recent large-scale efforts have demonstrated the potential of deep learning for materials discovery (Merchant et al., 2023). Periodic representations and boundary handling remain central challenges.

**Generative modeling on periodic manifolds.** Many approaches adapt diffusion to periodic variables by redefin-

ing noise processes or using manifold-specific distributions (Jiao et al., 2023; De Bortoli et al., 2022; Cornet et al., 2025). Flow-matching alternatives such as FlowMM (Miller et al., 2024) likewise need to handle the wrap-around geometry explicitly. While effective, these adaptations often introduce additional complexity in the forward process and the training objective. Bayesian Flow Networks (Graves et al., 2023) provide a practical advantage that the network operates on belief parameters rather than raw noisy samples, which can reduce the effective complexity of the denoising mapping. PeriodicBFN inherits this advantage while restoring the additive-accuracy structure that prior circular BFN adaptations (Wu et al., 2025) lack.

**Phase retrieval and compressive light fields.** Multi-layer phase modulation for 3D displays is commonly approached by tensor-display formulations and iterative physical or neural solvers (Wetzstein et al., 2012; Maruyama et al., 2020). Recent approaches also use neural regression models for real-time phase synthesis (Ma et al., 2025). Casting phase synthesis as a generative modeling problem provides a complementary perspective that can better represent multimodality and physical constraints.

## 3. Background: Bayesian Flow Networks and Periodic Variables

### 3.1. Bayesian Flow Networks: generation and training

A Bayesian Flow Network (BFN) (Graves et al., 2023) treats data as distributions: a sample $\mathbf{x}$ is viewed as the limit of a low-entropy belief concentrated around $\mathbf{x}$. BFN generation starts from a high-entropy prior belief and repeatedly injects network-guided information to produce progressively lower-entropy beliefs. Training learns the network that reads the current belief state (the "stored information") and outputs a denoised target used to form observations.

### 3.2. Euclidean BFNs in $\mathbb{R}^D$

**Belief representation.** For Euclidean data $\mathbf{x} \in \mathbb{R}^D$, we represent a belief over possible values of $\mathbf{x}$ by an isotropic Gaussian,

$$p(\mathbf{x} \mid \boldsymbol{\mu}, \rho) = \mathcal{N}\left(\mathbf{x} \mid \boldsymbol{\mu}, \rho^{-1}I\right), \qquad (1)$$

where $\rho \in \mathbb{R}_+$ is a scalar accuracy shared across dimensions at time $t$. For a training datum, the corresponding target belief is the low-entropy limit of (1) centered at that datum, i.e., $\boldsymbol{\mu}$ equals the datum and $\rho \to \infty$.

**Generation via iterative Bayesian conditioning.** We take $t = 0$ to denote the initial high-entropy belief and $t = 1$ the final low-entropy belief. For sampling, choose a discrete grid $0 = t_0 < t_1 < \cdots < t_T = 1$ and update the belief state $(\boldsymbol{\mu}_i, \rho_i)$ along this grid:

1. **Network readout:** input the current belief state (typically $\boldsymbol{\mu}_i$ and $t_i$) and predict $\hat{\mathbf{x}}_i = \hat{\mathbf{x}}(\boldsymbol{\mu}_i, t_i)$.

2. **Observation:** form a guided noisy observation with step accuracy $\alpha_i$,

$$\mathbf{y}_i = \hat{\mathbf{x}}_i + \boldsymbol{\epsilon}_i, \qquad \boldsymbol{\epsilon}_i \sim \mathcal{N}(\mathbf{0}, \alpha_i^{-1}I). \qquad (2)$$

3. **Belief update:** update the belief parameters to reduce entropy.

With an isotropic Gaussian belief and isotropic observation noise, the Bayes update can be written in vector form as

$$\rho_{i+1} = \rho_i + \alpha_i, \qquad (3)$$

$$\boldsymbol{\mu}_{i+1} = \frac{\rho_i\,\boldsymbol{\mu}_i + \alpha_i\,\mathbf{y}_i}{\rho_i + \alpha_i}. \qquad (4)$$

Repeating (4) maps the high-entropy input belief to a low-entropy output belief; a final sample can be obtained from $p(\mathbf{x} \mid \boldsymbol{\mu}_T, \rho_T)$ (or by taking $\boldsymbol{\mu}_T$).

**Training states from additive accuracy.** During generation, the neural network is queried on intermediate belief states $\boldsymbol{\mu}_i$ produced by repeated applications of (4). Thus, before inference, training must expose the network to the kinds of states it will encounter along this generative trajectory and teach it to map each state $\boldsymbol{\mu}_t$ back to the corresponding clean datum. The key property that makes this possible is additive accuracy: the update $\rho \leftarrow \rho + \alpha$ in (4) makes the accumulated information deterministic. With a continuous schedule $\alpha(t)$,

$$\rho(t) = \rho_0 + \int_0^t \alpha(s)\,ds, \qquad (5)$$

so the entropy level, and therefore the distribution of intermediate belief states, is known at each time $t$. Following the original BFN construction (Graves et al., 2023), we choose a schedule that makes the belief entropy decrease linearly over $t$. This yields the continuous-time observation accuracy

$$\alpha(t) = -\frac{2\ln(\sigma_1)}{\sigma_1^{2t}}, \qquad (6)$$

and we define

$$\gamma(t) = 1 - \sigma_1^{2t}, \qquad (7)$$

where we set $\sigma_1 = \sqrt{10^{-3}}$ in all experiments. Under this schedule, additive accuracy yields the closed-form forward marginal

$$p_F(\boldsymbol{\mu}_t \mid \mathbf{x}, t) = \mathcal{N}\left(\boldsymbol{\mu}_t \mid \gamma(t)\mathbf{x},\ \gamma(t)\bigl(1 - \gamma(t)\bigr)I\right), \quad (8)$$

which is used to sample the training states that the network will later see during generation. This gives the weighted denoising objective

$$L^\infty(\mathbf{x}) = \mathbb{E}_{t \sim \mathcal{U}(0,1)}$$

$$\mathbb{E}_{\boldsymbol{\mu}_t \sim p_F(\boldsymbol{\mu}_t \mid \mathbf{x}, t)}\left[\frac{\alpha(t)}{2}\|\mathbf{x} - \hat{\mathbf{x}}(\boldsymbol{\mu}_t, t)\|_2^2\right]. \qquad (9)$$

*Remark* 3.1 (Scalar Bayes updates and neural aggregation). Although (4) is written in vector form, it decomposes into independent scalar updates because the covariance is proportional to $I$. In other words, the Bayesian update only performs *per-dimension information accumulation* (each coordinate mean is updated using its own observation), while all multi-dimensional information fusion—e.g., how one coordinate should influence the denoised prediction of another—is handled by the neural network through the mapping $(\boldsymbol{\mu}_t, t) \mapsto \hat{\mathbf{x}}(\boldsymbol{\mu}_t, t)$. This separation between scalar Bayes updates and neural aggregation is a key design property that we will leverage in Section 4.

### 3.3. Periodic data: limitations of direct circular updates

For periodic data such as angles $\theta \in [0, 2\pi)$, one may try to keep the belief on the circle (e.g., a von Mises family (Mardia & Jupp, 1999)). The issue is that the circular posterior update does *not* generally admit an additive-accuracy form analogous to $\rho \leftarrow \rho + \alpha$ in (4). Consequently, $\rho(t) = \rho_0 + \int_0^t \alpha(s)ds$ no longer describes the information accumulation, so the forward marginal $p_F(\boldsymbol{\mu}_t \mid \mathbf{x}; t)$ used in (9) is not available in closed form. This is the source of the instability/schedule sensitivity of direct circular BFNs. We will formalize this limitation in Theorem 4.1; the same issue is also noted in prior periodic BFN work CrysBFN (Wu et al., 2025).

### 3.4. Applications of periodic data

#### 3.4.1. CRYSTAL STRUCTURE PREDICTION (CSP)

We represent a crystal by a triplet $M = (A, F, L)$. Let $N$ be the number of atoms and $K$ the number of element types. We denote atom types by $A \in \{1, \ldots, K\}^N$ (equivalently, a one-hot encoding $\tilde{A} \in \{0, 1\}^{N \times K}$). In the neural backbone, $A$ is embedded into per-atom features $E(A) \in \mathbb{R}^{N \times d_A}$. The periodic part is the fractional coordinates $F \in [0, 1)^{N \times 3}$, where the $i$-th row $\mathbf{f}_i \in [0, 1)^3$ specifies the position of atom $i$ in the unit cell. We parameterize the lattice by a matrix $L \in \mathbb{R}^{3 \times 3}$ whose rows are the lattice vectors. The corresponding Cartesian coordinates are given by $\mathbf{r}_i = \mathbf{f}_i L \in \mathbb{R}^{1 \times 3}$.

**Task formulation.** Crystal structure prediction (CSP) is posed as conditional generation of lattice and coordinates given composition:

$$p(L, F \mid A). \tag{10}$$

The training data is typically sourced from large-scale materials databases such as the Materials Project (Jain et al., 2013). The key challenge is that $F$ lives on a 3-torus due to periodic wrap-around; i.e., $F$ and $(F + \mathbf{1}) \bmod 1$ represent the same crystal. Moreover, $F$ is invariant to a global periodic translation, i.e., $F$ and $(F + \mathbf{t}) \bmod 1$ for any $\mathbf{t} \in [0, 1)^3$ describe the same structure. Our main focus in

this work is the design of a BFN that models such periodic coordinates $F$ while preserving additive accuracy.

#### 3.4.2. GLASSES-FREE 3D DISPLAYS VIA POSE-TRACKED COMPRESSIVE LIGHT FIELDS

We focus on glasses-free 3D displays implemented by pose-tracked compressive light fields (CLF), represented by the recent EyeReal system (Ma et al., 2025). At a high level, the display synthesizes two view-dependent images for the viewer's left/right eyes while tracking the eye pose in real time. The core computational problem is to infer phase patterns on a small number of cascaded LCD layers so that, after wave propagation and polarization optics, the emitted binocular light field matches a desired target.

**From a single ray to a global operator.** We consider a polarized-type CLF architecture with three cascaded LCD layers. On a single light ray, the polarization analyzer converts the accumulated phase modulation into an emitted intensity via Malus's law:

$$L = \sin^2\left(\sum_{n=1}^{3} \theta_n\right), \tag{11}$$

where $\theta_n$ is the phase contributed by the $n$-th LCD layer along that ray. Let $w \times h$ be the spatial resolution, and let $x \in \mathbb{R}^{3wh}$ stack all per-pixel phase parameters from the three layers. For a given viewer pose, inverse ray tracing determines, for each pixel in each eye view, which three LCD pixels (one per layer) the corresponding ray intersects. This induces a sparse linear operator $P(\text{pose}) \in \mathbb{R}^{2wh \times 3wh}$ (each row has at most three non-zeros) such that the ray-wise phase accumulation is written as $\sum_{n=1}^{3} \theta_n = (P(\text{pose})x)_m$. Stacking all rays from two viewpoints yields the vectorized forward model (with elementwise $\sin^2$)

$$\hat{b}(x; \text{pose}) = \sin^2(P(\text{pose})x), \tag{12}$$

where $\hat{b} \in \mathbb{R}^{2wh}$ denotes the rendered binocular intensities.

**Phase synthesis as a large-scale nonlinear problem.** Given a target binocular light field $b \in \mathbb{R}^{2wh}$, phase synthesis solves a constrained nonlinear least-squares problem

$$x^* = \arg\min_{x \in [0, \pi/2]^{3wh}} \left\|\sin^2(P(\text{pose})x) - b\right\|_2^2. \tag{13}$$

which is large-scale (on the order of $wh$ variables/residuals) and nonconvex. Physics-based iterative methods (e.g., conjugate-gradient-style solvers applied to suitable linearizations) can be too slow to meet the strict real-time latency required by glasses-free 3D displays.

**Why a generative view helps.** EyeReal demonstrates that real-time phase synthesis is possible by amortizing inference with a neural network, i.e., predicting phases directly

as $x = \text{net}(b, \text{pose})$ (Ma et al., 2025). However, there remains substantial room to improve display quality, e.g., view crosstalk caused by lost high-frequency details. We argue that a key missing ingredient is the periodic nature of phase variables (defined modulo $2\pi$, and effectively modulo $\pi$ under $\sin^2$), which suggests treating phase recovery as conditional generation rather than point estimation. Accordingly, we cast CLF phase synthesis as learning a conditional generative model

$$p(x \mid b, \text{pose}). \tag{14}$$

## 4. Method: PeriodicBFN

An overview of the full model pipeline is provided in Figure 2.

### 4.1. Why periodic data cannot be modeled by "staying on the circle"

Periodic variables (fractional coordinates modulo one, phases modulo $2\pi$) live on a torus rather than $\mathbb{R}^D$. A tempting approach is therefore to represent each periodic scalar with a circular distribution (e.g., von Mises) and perform Bayes updates directly on $\mathbb{S}^1$. However, this breaks the core BFN property in Section 3: the existence of a deterministic, additive accuracy state $\rho(t)$.

**Proposition 4.1** (No additive accuracy on the circle). *Consider any BFN whose per-step belief update is performed in a distribution family supported on the circle $\mathbb{S}^1$. In general, there does not exist a scalar "accuracy" parameter $\rho$ and per-step "observation accuracy" $\alpha$ such that the posterior update satisfies an additive law $\rho \leftarrow \rho + \alpha$ independent of the current mean direction.*

*Consequently, the accumulated information cannot be expressed as a deterministic function of time $t$ as in (5).*

Theorem 4.1 has three immediate consequences. First, the model must expose the current circular concentration/uncertainty to the network (breaking the clean interface where $t$ alone encodes information level). Second, one loses the closed-form entropy schedule $\alpha(t)$ that underpins jumpable training in (9). Third, without an analytic forward marginal analogous to (8), continuous-time training is unavailable and performance becomes sensitive to the discrete inference schedule.

### 4.2. Additive-accuracy embedding via Gaussian Bayes in $\mathbb{R}^2$

**Unit-circle embedding.** For a periodic scalar $f \in [0, 1)$, define its unit-circle embedding

$$\mathbf{u}(f) = \begin{bmatrix} \cos(2\pi f) \\ \sin(2\pi f) \end{bmatrix} \in \mathbb{R}^2. \tag{15}$$

We maintain a Gaussian belief over $\mathbf{u}(f)$ exactly in the form of (1):

$$p(\mathbf{u} \mid \boldsymbol{\mu}, \rho) = \mathcal{N}(\mathbf{u} \mid \boldsymbol{\mu}, \rho^{-1} I_2), \qquad \boldsymbol{\mu} \in \mathbb{R}^2, \ \rho \in \mathbb{R}_+. \tag{16}$$

Because the belief family is Gaussian, the Bayes update is the same additive-accuracy update as (4), now in two dimensions. Thus $\rho(t)$ remains deterministic and the same schedule machinery in Section 3 applies verbatim. This also follows the design principle in the background remark on scalar Bayes updates and neural aggregation: the Bayesian update only performs per-coordinate information accumulation, while the network is responsible for fusing multi-dimensional context when mapping $(\boldsymbol{\mu}_t, t) \mapsto \hat{\mathbf{u}}(\boldsymbol{\mu}_t, t)$.

**Forward marginal and continuous-time objective.** We reuse the background schedule

$$\gamma(t) := 1 - \sigma_1^{2t}, \tag{17}$$

and sample intermediate training states from the embedded analogue of (8). Let $\mathbf{u}_f = \mathbf{u}(f)$; then

$$p_F(\boldsymbol{\mu}_t \mid \mathbf{u}_f, t) = \mathcal{N}(\boldsymbol{\mu}_t \mid \gamma(t)\mathbf{u}_f, \ \gamma(t)(1 - \gamma(t))I_2). \tag{18}$$

The network reads $(\boldsymbol{\mu}_t, t)$ and predicts a denoised unit vector $\hat{\mathbf{u}}(\boldsymbol{\mu}_t, t)$. We train with the same BFN weighting as (9):

$$L_{\text{per}}^\infty(f) = \mathbb{E}_{t \sim \mathcal{U}(0,1)} \mathbb{E}_{\boldsymbol{\mu}_t \sim p_F(\cdot \mid \mathbf{u}(f), t)} \left[ \frac{\alpha(t)}{2} \|\mathbf{u}(f) - \hat{\mathbf{u}}(\boldsymbol{\mu}_t, t)\|_2^2 \right]. \tag{19}$$

**On-circle parametrization (head).** To keep the denoised prediction on the unit circle away from the initial prior state while preserving the Euclidean BFN interface, we parameterize

$$\hat{\mathbf{u}}(\boldsymbol{\mu}_t, t) = \phi(x, y, t, \gamma),$$

$$\phi(x, y, t, \gamma) := \begin{cases} \mathbf{0}, & t < t_{\min}, \\ (\cos(\hat{\epsilon}), \ \sin(\hat{\epsilon})), & \text{otherwise,} \end{cases} \tag{20}$$

where $(x, y) = \boldsymbol{\mu}_t$, $t_{\min} = 10^{-10}$, and $\hat{\epsilon} = \text{net}(x, y, t, \gamma)$ is the raw network output. Note that, except at the initial prior state, we constrain the denoised prediction $\hat{\mathbf{u}}$ to lie on $\mathbb{S}^1$, while the noisy state $\boldsymbol{\mu}_t$ is unconstrained and can vary in radius. This is not an arbitrary extra degree of freedom: the radial magnitude is a natural carrier of uncertainty under the Gaussian belief parameterized by $\rho(t)$. In this sense, the 2D embedding makes explicit an auxiliary signal that many practical generators already use implicitly (e.g., accumulated accuracy in CrysBFN (Wu et al., 2025) and velocity in KLDM (Cornet et al., 2025)), while preserving the strictly additive-accuracy schedules required by jumpable BFN training.

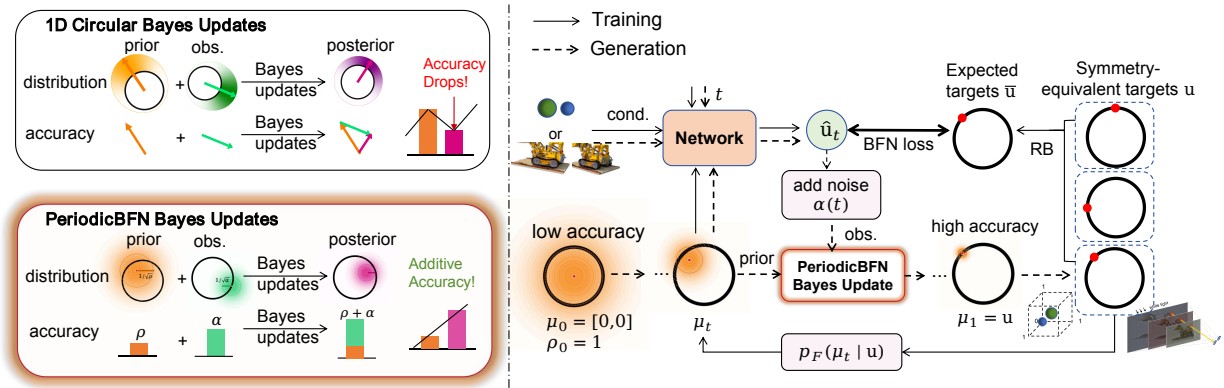

*Figure 2.* Overall framework of PeriodicBFN. **(Left) Accuracy accumulation.** Our 2D embedding decouples the spatial mean from scalar precision, guaranteeing strictly additive accuracy without information loss. **(Right) Pipeline.** Belief states evolve from a high-entropy prior at the origin toward a low-entropy state on the unit circle. Invariant Rao–Blackwellized (RB) targets resolve nuisance symmetries by analytically marginalizing equivalent data into a low-variance expected center of mass.

Beyond a positional-encoding convenience, the 2D embedding is what enables Theorem 4.1 to be bypassed: Gaussian Bayes updates in Cartesian $\mathbb{R}^2$ restore strict additive accuracy and an exact closed-form forward marginal (18), and also make the closed-form Rao–Blackwellized objective in Section 4.5 tractable.

### 4.3. Generation algorithm: discrete Gaussian Bayes updates

At inference, we follow the standard discrete BFN update rule, now applied to the embedded state in $\mathbb{R}^2$. We use $t = 0$ for the initial high-entropy state and $t = 1$ for the final generated state. For $T$ update steps, define $t_i = i/T$ for $i = 0, \ldots, T$ and use the step accuracies

$$\alpha_i := \sigma_1^{-2(i+1)/T}\left(1-\sigma_1^{2/T}\right), \qquad i = 0, \ldots, T-1. \quad (21)$$

Starting from $(x_0, y_0) = (0, 0)$ and $\rho_0 = 1$, each step predicts a denoised unit vector at $t_i$, samples an observation $(\tilde{x}_i, \tilde{y}_i)$, and applies the Gaussian Bayes update:

$$\rho_{i+1} = \rho_i + \alpha_i,$$
$$(x_{i+1}, y_{i+1}) = \frac{\rho_i(x_i, y_i) + \alpha_i(\tilde{x}_i, \tilde{y}_i)}{\rho_{i+1}}. \quad (22)$$

Finally we output $\hat{\mathbf{u}}_{\text{out}} = \phi(x_T, y_T, 1, 1 - \sigma_1^2)$ and recover $f$ from its angle.

### 4.4. Invariances in CSP and CLF

Periodic variables often come with nuisance symmetries. If the training target breaks these symmetries while inference enforces them, the objective becomes mismatched and gradients acquire avoidable variance. We therefore make the invariances explicit and incorporate them into the training target.

---

**Algorithm 1** PeriodicBFN sampling for one periodic scalar

**Require:** steps $T$, schedule $\{\alpha_i\}_{i=0}^{T-1}$, prior $\rho_0 \leftarrow 1$, state $(x_0, y_0) \leftarrow (0, 0)$
1: **for** $i = 0$ to $T - 1$ **do**
2: $\quad t_i \leftarrow i/T, \gamma_i \leftarrow \gamma(t_i)$
3: $\quad (\hat{x}_i, \hat{y}_i) \leftarrow \phi(x_i, y_i, t_i, \gamma_i)$
4: $\quad$ sample $(\tilde{x}_i, \tilde{y}_i) \sim \mathcal{N}((\hat{x}_i, \hat{y}_i), \alpha_i^{-1}I_2)$
5: $\quad \rho_{i+1} \leftarrow \rho_i + \alpha_i$
6: $\quad (x_{i+1}, y_{i+1}) \leftarrow \left(\rho_i(x_i, y_i) + \alpha_i(\tilde{x}_i, \tilde{y}_i)\right)/\rho_{i+1}$
7: **end for**
8: $(\hat{x}, \hat{y}) \leftarrow \phi(x_T, y_T, 1, 1 - \sigma_1^2)$
9: **return** $\hat{f} \leftarrow \frac{1}{2\pi}\text{atan2}(\hat{y}, \hat{x}) \bmod 1$

---

**Crystal structure prediction (CSP): periodic translation invariance.** For fractional coordinates $F \in [0, 1)^{N \times 3}$, a global periodic translation $F \mapsto (F + \mathbf{t}) \bmod 1$ with $\mathbf{t} \in [0, 1)^3$ leaves the crystal unchanged. In the unit-circle embedding, this corresponds to one unknown global rotation per fractional-coordinate axis, shared by all atoms on that axis.

**Compressive light-field (CLF): $\sin^2$ symmetry (within-period sign equivalence).** The forward model (12) depends on phases only through $\sin^2(\cdot)$. Consequently, at the ray level one has the invariance

$$\sin^2(\vartheta) = \sin^2(-\vartheta) = \sin^2(\vartheta + \pi), \quad (23)$$

which makes the inverse problem intrinsically ambiguous within each $\pi$-period. When training a generative model for phases, the target should respect this equivalence to avoid penalizing equally valid solutions.

## 4.5. Rao–Blackwellized targets and training losses

We apply Rao–Blackwellization (Robert & Roberts, 2021): whenever a nuisance symmetry variable is latent, we replace the random (symmetry-dependent) target by its conditional expectation. This yields an explicitly invariant objective with reduced gradient variance. We present the two cases corresponding to CSP and CLF.

**Intuition and guarantees.** Both CSP and CLF involve a latent nuisance symmetry (a continuous global periodic shift; a binary $\pi$-shift / sign equivalence). A naive denoising target depends on a sampled value of this symmetry even though all such targets describe the same physical configuration; Rao–Blackwellization replaces a single noisy realization by its conditional expectation. This is conceptually analogous to the equivariant denoising target in CSP diffusion (Lin et al., 2024), but the averaging is done analytically, yielding two strict guarantees: (i) *exact forward marginals* from additive accuracy via (18), and (ii) *guaranteed gradient-variance reduction* by the Rao–Blackwell theorem on the targets in (26) (and its CLF analogue).

### 4.5.1. CSP: MARGINALIZING A CONTINUOUS GLOBAL SHIFT

We present the $N$-atom case for one fractional-coordinate axis; the three-axis extension applies the same construction independently to each axis. Let the latent global periodic translation be $\delta \sim \mathrm{Unif}(0,1)$. Denote the ground-truth fractional coordinate along this axis by $f_i \in [0,1)$ for atom $i$, and recall the unit-circle embedding $\mathbf{u}(f) = (\cos(2\pi f), \sin(2\pi f))$. For known $\gamma \in (0,1)$, the embedded forward marginal implies the observation model

$$(x_i, y_i) \sim \mathcal{N}\Big(\gamma\,\mathbf{u}(f_i + \delta),\ \gamma(1-\gamma)I_2\Big), \qquad i = 1, \ldots, N, \tag{24}$$

where we implicitly take $f_i + \delta$ modulo 1 (equivalently, use the periodicity of $\mathbf{u}$). Using complex notation $z_i = x_i + jy_i$ for the embedded noisy state of atom $i$ and denoting $\mathbf{z} := (z_1, \ldots, z_N)$, the posterior over $\delta$ is von Mises with sufficient statistic

$$S := \sum_{i=1}^{N} z_i e^{-j2\pi f_i}, \qquad \psi := \arg S, \qquad \kappa := \frac{|S|}{1-\gamma}. \tag{25}$$

Let $A_n(\kappa) = I_n(\kappa)/I_0(\kappa)$. Then the Rao–Blackwellized (shift-marginalized) target for each atom on this axis is

$$\bar{\mathbf{u}}_i := \mathbb{E}[\mathbf{u}((f_i + \delta) \bmod 1) \mid \mathbf{z}]$$
$$= A_1(\kappa) \begin{bmatrix} \cos(2\pi f_i + \psi) \\ \sin(2\pi f_i + \psi) \end{bmatrix}. \tag{26}$$

**CSP loss.** Let $\hat{\mathbf{u}}_i(\boldsymbol{\mu}_t, t)$ be the model prediction (via (20)) for atom $i$ on this axis. The corresponding invariant training objective follows (19) with the RB target in (26):

$$L_{\mathrm{RB\text{-}CSP}}^{\infty} = \mathbb{E}_{t, \mathbf{z}}\left[\frac{\alpha(t)}{2} \sum_i \|\hat{\mathbf{u}}_i(\boldsymbol{\mu}_t, t) - \bar{\mathbf{u}}_i\|_2^2\right], \tag{27}$$

where $t \sim \mathcal{U}(0,1)$, $\mathbf{z}$ is sampled from (24), and the sum runs over the $N$ atoms sharing the same global translation $\delta$. Thus (26) is the posterior mean of the globally shifted circular embedding, replacing a noisy shift-dependent target by its expectation under $p(\delta \mid \mathbf{z})$. This is analogous to subtracting a circular center of mass in equivariant CSP pipelines (Lin et al., 2024), but the 2D embedding gives a closed form via von Mises moments.

### 4.5.2. CLF: DENOISING IN THE PHASE-ACCUMULATION ($Px$) SPACE

In CLF, we do not have ground-truth phase patterns $x$ for the three LCD layers; supervision is naturally defined in the measurement space through the differentiable renderer in (12). Let $\mathbf{y} := P(\mathrm{pose})\,x \in \mathbb{R}^{2wh}$ denote the ray-wise phase accumulation; since $\sin^2$ is $\pi$-periodic, $\mathbf{y}$ is only identifiable modulo $\pi$. We therefore apply PeriodicBFN to each scalar component $y_m$ in the $Px$ space rather than to $x$ directly, while still using $x$ as a feasibility-respecting parameterization of $\mathbf{y}$.

**$\pi$-periodic embedding with observable target.** Define a $\pi$-periodic unit-circle embedding

$$\mathbf{v}(y) := \begin{bmatrix} \cos(2y) \\ \sin(2y) \end{bmatrix} \in \mathbb{R}^2. \tag{28}$$

From $b_m = \sin^2(y_m)$ the cosine component is recovered exactly,

$$\cos(2y_m) = 1 - 2b_m, \tag{29}$$

while $\sin(2y_m)$ is ambiguous up to a binary sign $s_m \in \{\pm 1\}$, giving the two consistent embedded targets

$$\mathbf{v}(b_m, s_m) := \begin{bmatrix} 1 - 2b_m \\ s_m\, 2\sqrt{b_m(1-b_m)} \end{bmatrix}. \tag{30}$$

**Rao–Blackwellized target and CLF loss.** Let $\mathbf{m} = (m^{\cos}, m^{\sin}) \in \mathbb{R}^2$ be the noisy training state at time $t$ for a given ray and write $\gamma = \gamma(t)$. With $\ell(b_m) := 2\sqrt{b_m(1-b_m)}$, marginalizing the binary sign latent $s_m$ analytically against its Bernoulli posterior under the embedded forward marginal yields a closed-form Rao–Blackwellized target (full derivation in Section A.4):

$$\bar{\mathbf{v}}_{\mathrm{CLF}}(\mathbf{m}, b_m) = \begin{bmatrix} 1 - 2b_m \\ \ell(b_m)\, \tanh\left(\dfrac{m^{\sin}\,\ell(b_m)}{1-\gamma}\right) \end{bmatrix}. \tag{31}$$

The corresponding invariant CLF objective mirrors (19), applying this RB target elementwise over rays:

$$L_{\text{RB-CLF}}^{\infty} = \mathbb{E}_{t,\,\mathbf{m}} \left[ \frac{\alpha(t)}{2} \left\| \hat{\mathbf{v}}(\mathbf{m}, t) - \bar{\mathbf{v}}_{\text{CLF}}(\mathbf{m}, b) \right\|_2^2 \right],$$
(32)

where $\hat{\mathbf{v}}(\mathbf{m}, t)$ is the network prediction in the embedded $Px$ space. This loss respects the $\pi$-period sign equivalence implied by $\sin^2$ and avoids choosing an arbitrary representative during training. In implementation we augment (32) with a small renderer-aligned reconstruction term for optimization stability; the practical hybrid objective and the inference-time sampling procedure are deferred to Section A.5.

### 4.6. Architecture of CSP and CLF

For both tasks we reuse the backbone of the strongest prior generative method as is and only adapt the periodic output parameterization required by PeriodicBFN.

**CSP architecture.** We reuse the CrysBFN equivariant backbone (Wu et al., 2025) for crystals, which builds on E(3)-equivariant graph neural networks. Lattice parameters $L$ are generated with the standard Euclidean BFN head, while fractional coordinates $F$ use the periodic embedding head in (20). Both heads share the same time variable $t$ and schedule, so the model learns joint lattice–fractional geometry consistently.

**CLF architecture.** We reuse the U-Net backbone (Ronneberger et al., 2015) of the EyeReal phase-synthesis pipeline (Ma et al., 2025) unchanged, and adapt only its input and output channel counts to host the 2D embedded belief required by PeriodicBFN. The network produces a base LCD phase pattern together with additive residual modulators on the cosine and sine components of the embedded target. Full details of the input/output layout, time conditioning, the output head parameterization, and the design rationale are provided in Section A.5.

## 5. Experiments

### 5.1. Crystal Structure Prediction

We target stable structure prediction $p(L, F \mid A)$, with atom types $A$ injected into the network by concatenating node features with atom-type embeddings, following (Jiao et al., 2023).

**Baselines and metrics.** We compare against diffusion-based **CD-VAE** (Xie et al., 2022), **DiffCSP** (Jiao et al., 2023), **EquiCSP** (Lin et al., 2024), the flow-matching method **FlowMM** (Miller et al., 2024), and the BFN-based **CrysBFN** (Wu et al., 2025). We report **Match Rate** and

**RMSE** computed by pymatgen's `StructureMatcher` (`stol=0.5`, `angle_tol=10`, `ltol=0.3`) (Ong et al., 2013).

**Results.** Table 1 shows PeriodicBFN consistently improves over baselines, with +18.6 match-rate points over DiffCSP on MP-20 and +5.7 / +6.5 points over the strongest baseline (CrysBFN) on MP-20 / MPTS-52. The only metric where PeriodicBFN trails marginally is MPTS-52 RMSE (0.1105 vs. 0.1038); the multi-seed evaluation in Section B shows this single-seed gap is not statistically meaningful and PeriodicBFN in fact has a slightly lower mean with much smaller variance ($0.1105 \pm 0.0024$ vs. $0.1112 \pm 0.0077$).

**Ablations on MP-20.** Table 2 ablates the Rao–Blackwellized objective and the 2D unit-circle embedding; both are needed for the full performance.

PeriodicBFN is also markedly more robust to the Number of Function Evaluations (NFE) than the 1D circular baseline, reaching its peak match rate already at 200–500 forwards while CrysBFN's accuracy degrades when its discrete schedule is shortened (full sweep in Section E). This empirically validates the theoretical motivation behind our additive-accuracy formulation: because the embedded forward marginal (18) is exact in continuous time, sampling stays stable as the step count varies, whereas CrysBFN's reliance on hand-tuned discrete schedules makes it brittle outside the schedule it was trained for.

### 5.2. Compressive Light-Field Phase Synthesis

We test PeriodicBFN on a multi-layer phase synthesis task for compressive light-field displays.

Our CLF data is generated using the open-source EyeReal pipeline (Ma et al., 2025) with the `lego_bulldozer` NeRF 3D scene: we render 10,000 target binocular pairs for training by randomly sampling viewer poses and binocular distance $R \in [30, 50]\,\text{cm}$, and a separate held-out set of 10 pairs drawn from a disjoint pose distribution as an out-of-distribution (OOD) test split. Reconstruction quality (PSNR) is reported on the OOD test split at a resolution of $360{\times}640$, while end-to-end inference latency (Time / FPS) is measured with the same network architecture and parameters at the EyeReal display resolution of $1080{\times}1920$ to reflect deployment-relevant cost. The fully convolutional U-Net backbone makes the architecture agnostic to spatial resolution; only the input and output tensor sizes change between the two measurements.

As shown in Table 3 and Figure 3, EyeReal + PeriodicBFN improves reconstruction quality over EyeReal while remaining real-time, and it reduces left–right view crosstalk in challenging regions.

*Table 1.* Comparison on MP-20 and MPTS-52. Best per column in **bold**. Multi-seed mean±std results are reported in Section B.

| | MP-20 | | MPTS-52 | |
|---|---|---|---|---|
| | Match rate↑ | RMSE↓ | Match rate↑ | RMSE↓ |
| CDVAE (Xie et al., 2022) | 33.90 | 0.1045 | 5.34 | 0.2106 |
| DiffCSP (Jiao et al., 2023) | 51.49 | 0.0631 | 12.19 | 0.1786 |
| EquiCSP (Lin et al., 2024) | 57.59 | 0.0510 | 14.85 | 0.1169 |
| FlowMM (Miller et al., 2024) | 61.39 | 0.0566 | 17.54 | 0.1726 |
| CrysBFN (Wu et al., 2025) | 64.35 | 0.0433 | 20.52 | **0.1038** |
| PeriodicBFN | **70.05** | **0.0370** | **27.03** | 0.1105 |

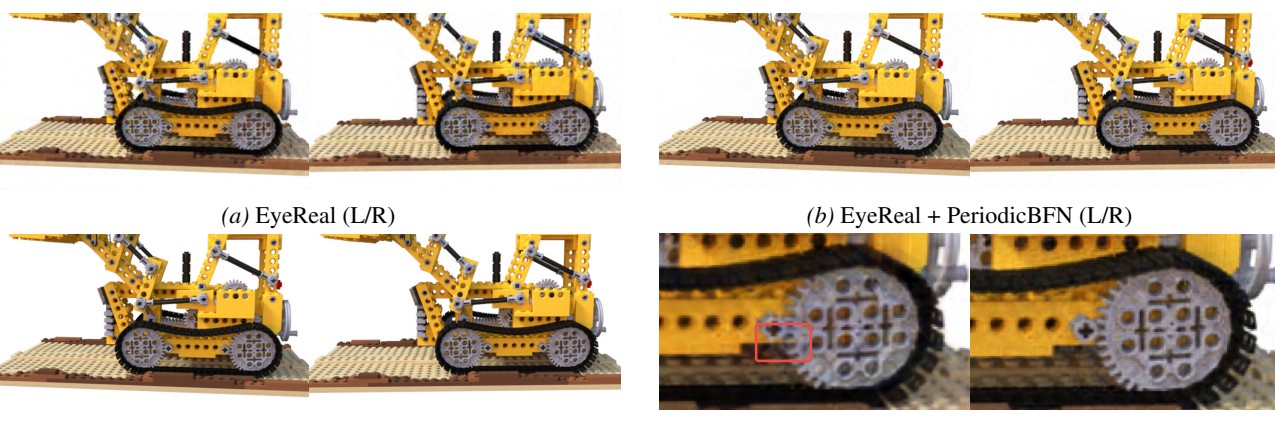

*(a)* EyeReal (L/R)

*(b)* EyeReal + PeriodicBFN (L/R)

*(c)* GT (L/R)

*(d)* Zoom-in (EyeReal vs. ours)

*Figure 3.* Qualitative CLF phase-synthesis comparison. EyeReal exhibits left–right view crosstalk (leakage between views), which is alleviated by EyeReal + PeriodicBFN; the zoom-in highlights the difference.

*Table 2.* MP-20 ablations (RB = Rao–Blackwellization; $t$ input = scalar time/accuracy signal). Multi-seed mean±std results in Section B.

| Method | Match rate (%)↑ | RMSE↓ |
|---|---|---|
| PeriodicBFN | 70.05 | 0.0370 |
| w/o RB | 66.11 | 0.0507 |
| w/o 2D emb. | 60.02 | 0.0569 |
| w/o $t$ input | 66.17 | 0.0479 |

*Table 3.* CLF phase synthesis results (end-to-end FPS). Underline = second-best per column. Multi-seed mean±std results are reported in Section B.

| Method | PSNR↑ | Time (ms)↓ | FPS↑ |
|---|---|---|---|
| Conj. Grad. | 35.51 | 692.0 | 1.4 |
| EyeReal | 28.86 | 19.9 | 52.1 |
| EyeReal + PeriodicBFN | 31.40 | 35.1 | 28.49 |

## 6. Conclusion

We presented PeriodicBFN, a principled periodic extension of Bayesian Flow Networks that restores strictly additive accuracy by embedding periodic coordinates into a 2D unit-circle representation and performing Gaussian Bayesian updates in Cartesian space. We further derived a Rao–Blackwellized invariant loss that marginalizes global circular shifts in closed form. This yields a stable, modular framework for periodic generative modeling, spanning crystal structure prediction and compressive light-field phase synthesis, and the formulation extends naturally to other periodic variables such as molecular torsion angles (Jing et al., 2022) and periodic time-series. A practical limitation is that, while real-time, our iterative CLF inference is slower per frame than one-shot regressors; lighter periodic-aware backbones or distillation are natural directions for further acceleration.

## Acknowledgements

This work was supported by the Guangdong S&T Program under Grant No. 2024B0101040005, and the National Natural Science Foundation of China under Grant No. 625B2054. The authors thank the anonymous reviewers for their constructive feedback.

## Impact Statement

This work aims to improve the reliability of generative modeling for periodic scientific variables, with potential benefits for materials discovery and optical system design. As with other generative methods, downstream impacts depend on deployment: models could accelerate benign research

workflows, but may also be misused to generate misleading synthetic data if applied without domain validation. We do not introduce new data collection or user-facing decision systems; we encourage reporting uncertainty, enforcing physical constraints where applicable, and releasing evaluation protocols alongside any software.

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

# A. Additional Derivations and Details

## A.1. Geometric intuition

The key obstruction to additive accuracy on the circle is that a circular posterior update depends on the *relative angle* between the current mean direction and the new observation. Concretely, consider a von Mises prior over an angle $\theta$ with mean direction $\mu$ and concentration $\kappa$, and a von Mises "observation" with mean $\theta_{\text{obs}}$ and concentration $\alpha$. The posterior is again von Mises, but its concentration is the length of a vector sum: $\kappa' = \|\kappa\,\mathbf{u}(\mu) + \alpha\,\mathbf{u}(\theta_{\text{obs}})\|_2$. This depends on $\Delta := \theta_{\text{obs}} - \mu$ through $\kappa' = \sqrt{\kappa^2 + \alpha^2 + 2\kappa\alpha\cos\Delta}$, so there is no state-independent additive law of the form $\kappa' \leftarrow \kappa + \alpha$. As a result, the accumulated information level cannot be expressed as a deterministic schedule in time $t$, and the closed-form forward marginal used by jumpable BFNs is generally unavailable.

## A.2. Non-additivity of direct circular Bayesian updates

We give a simple proof sketch for the von Mises family. Let the prior be $p(\theta) \propto \exp\left(\kappa\cos(\theta - \mu)\right)$ and the likelihood be $p(\theta_{\text{obs}} \mid \theta) \propto \exp\left(\alpha\cos(\theta - \theta_{\text{obs}})\right)$. Using the unit-vector embedding $\mathbf{u}(\theta) = (\cos\theta, \sin\theta)$, we can rewrite the log-density of the posterior (up to constants) as

$$\kappa\,\mathbf{u}(\mu)^\top\mathbf{u}(\theta) + \alpha\,\mathbf{u}(\theta_{\text{obs}})^\top\mathbf{u}(\theta) = \left(\kappa\,\mathbf{u}(\mu) + \alpha\,\mathbf{u}(\theta_{\text{obs}})\right)^\top\mathbf{u}(\theta). \tag{33}$$

Therefore the posterior is von Mises with mean direction $\mu' = \arg\left(\kappa\mathbf{u}(\mu) + \alpha\mathbf{u}(\theta_{\text{obs}})\right)$ and concentration

$$\kappa' = \|\kappa\,\mathbf{u}(\mu) + \alpha\,\mathbf{u}(\theta_{\text{obs}})\|_2 = \sqrt{\kappa^2 + \alpha^2 + 2\kappa\alpha\cos(\theta_{\text{obs}} - \mu)}. \tag{34}$$

Since $\kappa'$ depends on $\mu$ via the relative angle, no update of the form $\kappa' \leftarrow \kappa + \alpha$ (independent of the current mean direction) can hold in general.

## A.3. Derivation of the Rao–Blackwellized objective for CSP

Starting from (24) with $z_i = x_i + jy_i$, the conditional density of $\mathbf{z}$ given $\delta$ is

$$p(\mathbf{z} \mid \delta) \propto \exp\left(-\frac{1}{2\gamma(1-\gamma)}\sum_{i=1}^{N}\|(x_i, y_i) - \gamma\,\mathbf{u}(f_i + \delta)\|_2^2\right). \tag{35}$$

Dropping terms independent of $\delta$ and collecting the $\delta$-dependent part yields

$$p(\delta \mid \mathbf{z}) \propto \exp\left(\frac{1}{1-\gamma}\,\Re\left(e^{j2\pi\delta}\,\overline{S}\right)\right), \tag{36}$$

where $S := \sum_{i=1}^{N} z_i e^{-j2\pi f_i}$. This is a von Mises distribution with mean angle $\psi = \arg S$ and concentration $\kappa = |S|/(1-\gamma)$ as in (25). Its first circular moment satisfies

$$\mathbb{E}\left[e^{j2\pi\delta} \mid \mathbf{z}\right] = A_1(\kappa)\,e^{j\psi}, \qquad A_1(\kappa) = I_1(\kappa)/I_0(\kappa). \tag{37}$$

Using $\mathbf{u}(f_i + \delta) = (\cos(2\pi(f_i + \delta)), \sin(2\pi(f_i + \delta)))$ gives

$$\bar{\mathbf{u}}_i = \mathbb{E}[\mathbf{u}(f_i + \delta) \mid \mathbf{z}] = A_1(\kappa)\begin{bmatrix}\cos(2\pi f_i + \psi)\\\sin(2\pi f_i + \psi)\end{bmatrix}, \tag{38}$$

which matches (26).

## A.4. Derivation of the Rao–Blackwellized objective for CLF

We derive the closed form (31) for the CLF Rao–Blackwellized target by marginalizing the binary sign latent $s_m \in \{\pm1\}$. Recall the two embedded targets and the Bernoulli posterior:

$$\mathbf{v}(b_m, s_m) = \begin{bmatrix}1 - 2b_m\\s_m\,\ell(b_m)\end{bmatrix}, \quad \ell(b_m) := 2\sqrt{b_m(1-b_m)}, \qquad p(s_m \mid \mathbf{m}, b_m) \propto \exp\left(\frac{\mathbf{m}^\top\mathbf{v}(b_m, s_m)}{1-\gamma}\right). \tag{39}$$

Writing $\mathbf{m} = (m^{\cos}, m^{\sin})$, the inner product splits as

$$\mathbf{m}^\top \mathbf{v}(b_m, \pm 1) = \underbrace{m^{\cos}(1 - 2b_m)}_{\text{sign-invariant}} \pm m^{\sin} \ell(b_m). \tag{40}$$

The first term is identical under $s_m = \pm 1$ and therefore cancels in the normalization of the Bernoulli posterior, leaving a one-dimensional logistic in the $\sin$-channel:

$$p(s_m = +1 \mid \mathbf{m}, b_m) = \sigma\left(\frac{2\, m^{\sin} \ell(b_m)}{1 - \gamma}\right), \qquad \sigma(x) := (1 + e^{-x})^{-1}. \tag{41}$$

Plugging these probabilities into the conditional mean

$$\bar{\mathbf{v}}_{\text{CLF}}(\mathbf{m}, b_m) = \sum_{s \in \{\pm 1\}} p(s \mid \mathbf{m}, b_m)\, \mathbf{v}(b_m, s), \tag{42}$$

the $\cos$-component evaluates to $1 - 2b_m$ (independent of $s_m$, with the Bernoulli probabilities summing to one), while the $\sin$-component evaluates to

$$\big(p(s_m = +1 \mid \mathbf{m}, b_m) - p(s_m = -1 \mid \mathbf{m}, b_m)\big) \ell(b_m) = \tanh\left(\frac{m^{\sin} \ell(b_m)}{1 - \gamma}\right) \ell(b_m), \tag{43}$$

where we used the identity $\sigma(2u) - \sigma(-2u) = \tanh(u)$. Combining the two components recovers the closed form in (31), which is directly computable from $b_m$, $\mathbf{m}$, and $\gamma(t)$ without any explicit Bernoulli marginalization.

## A.5. Practical implementation of the CLF pipeline

This section collects the implementation details deferred from Sections 4.5.2 and 4.6: the embedded Bernoulli posterior used in the RB derivation, the hybrid training objective, the sampling procedure, and the U-Net architecture with its embedded output head.

**Bernoulli posterior in the embedded space.** Assume the BFN forward marginal in the embedded space takes the same Gaussian form as (18). For a given ray, the noisy state $\mathbf{m} \in \mathbb{R}^2$ at time $t$ satisfies

$$\mathbf{m} \mid (b_m, s_m) \sim \mathcal{N}\big(\gamma\, \mathbf{v}(b_m, s_m),\ \gamma(1 - \gamma)I_2\big), \tag{44}$$

which yields a Bernoulli posterior over the sign latent

$$p(s_m \mid \mathbf{m}, b_m) \propto \exp\left(\frac{\mathbf{m}^\top \mathbf{v}(b_m, s_m)}{1 - \gamma}\right). \tag{45}$$

The closed-form RB target (31) is obtained by marginalizing $s_m$ against this posterior; see Section A.4.

**Hybrid training objective.** In implementation, the embedded RB target (32) is combined with a renderer-aligned reconstruction term using the supervised intensity $b \in [0, 1]^{2wh}$:

$$L_{\text{CLF}}^{\text{hybrid}} = \mathbb{E}_{t,\,\mathbf{m}} \left[\frac{\alpha(t)}{2}\Big(\big\|\hat{b}\big(\hat{\mathbf{v}}(\mathbf{m}, t)\big) - b\big\|_2^2 + \lambda \left\|\hat{\mathbf{v}}(\mathbf{m}, t) - \bar{\mathbf{v}}_{\text{CLF}}(\mathbf{m}, b)\right\|_2^2\Big)\right], \tag{46}$$

where the rendered intensity is decoded from the cosine component via $\hat{b}(\hat{\mathbf{v}}) = (1 - \hat{v}^{\cos})/2$ ((29)), and the shared weight $\alpha(t)$ keeps both terms on the BFN entropy-rate schedule (6). The reconstruction term supplies a direct gradient along the differentiable renderer (12), helpful in the strongly underdetermined three-layer CLF inverse problem, while the embedded RB term enforces the periodic Bayes-optimal target (31). Both terms respect the $\pi$-period sign equivalence: the reconstruction term acts entirely through $\hat{v}^{\cos}$, which is uniquely determined by $b$ via (29), and the RB term marginalizes the sign latent by construction. We use $\lambda = 10^{-2}$ throughout.

**Sampling in the embedded $Px$ space.** At inference, we initialize the embedded belief at the origin $\mathbf{m}^{(0)} = \mathbf{0}$ and apply the standard BFN end-back update (Graves et al., 2023) on a schedule $0 = t_0 < t_1 < \cdots < t_N = 1$:

$$\mathbf{m}^{(i)} \;=\; \gamma_{i-1}\,\hat{\mathbf{v}}\big(\mathbf{m}^{(i-1)}, t_{i-1}\big) \;+\; \sqrt{\gamma_{i-1}\big(1 - \gamma_{i-1}\big)}\,\boldsymbol{\xi}_i, \qquad \boldsymbol{\xi}_i \sim \mathcal{N}(\mathbf{0}, I_2), \tag{47}$$

with $\gamma_i = \gamma(t_i)$. The convention follows the resample-at-left-endpoint variant of end-back: at iteration $i$ the belief is resampled from the forward marginal at $t_{i-1}$ using the network's prediction at the same time; the first iteration at $t_0 = 0$ (with $\gamma_0 = 0$) amounts to a pure prior draw and is effectively a warm-up, while a final forward at $t_N = 1$ yields $\hat{\mathbf{v}}(\mathbf{m}^{(N)}, 1)$, from which the displayed LCD patterns are read off (via the output head described below) and the rendered intensity is obtained as $\hat{b} = (1 - \hat{v}^{\cos})/2$. Since the embedded forward marginal underlying (44) is exact in continuous time, the procedure is robust to the schedule and the number of steps; in our CLF experiments a small number of steps ($N \in \{1, 2\}$) already matches the quality of much longer chains.

**U-Net backbone and conditioning.** We reuse the U-Net backbone (Ronneberger et al., 2015) of the EyeReal phase-synthesis pipeline (Ma et al., 2025) unchanged, and adapt only its input and output channel counts. The network operates in screen space (one tensor per LCD layer), with the warp/unwarp operators inherited from EyeReal implementing the rasterization induced by $P(\text{pose})$ and its adjoint, so the U-Net sees screen-space tensors while supervision is naturally defined in the ray-accumulation domain $\mathbf{y} = P(\text{pose})x$. Conditioning on $(b, \text{pose})$ is inherited from EyeReal: the target binocular intensity $b$ is warped to screen space and concatenated to the U-Net input, alongside the cosine and sine components of the current embedded belief $(\mathbf{m}^{\cos}, \mathbf{m}^{\sin})$ likewise warped to screen space. Time conditioning is injected through adaptive group normalization—a standard practice in diffusion- and BFN-style backbones: a sinusoidal embedding of $t$ is mapped through a two-layer MLP to channel-wise affine parameters $(\boldsymbol{\eta}_t, \boldsymbol{\beta}_t)$ that modulate the GroupNorm features at each U-Net scale. We use a distinct symbol $\boldsymbol{\eta}_t$ here to avoid notational collision with the BFN noise schedule $\gamma(t)$.

**Embedded output head with residual cosine/sine modulators.** The U-Net produces three output heads in screen space, each yielding one quantity per LCD layer (per RGB channel). The first head outputs a base LCD phase pattern $x_{\text{base}}$ (passed through a $\sigma(\cdot) - 0.2$ activation that mildly constrains the per-layer range, consistent with EyeReal's parameterization). The remaining two heads output per-pixel additive modulators $(\boldsymbol{\delta}^{\cos}, \boldsymbol{\delta}^{\sin})$ on the cosine and sine components of the embedding. All three quantities are produced by a single U-Net forward pass and therefore depend implicitly on $(\mathbf{m}, t, b, \text{pose})$; we leave this dependence inside the U-Net for brevity. After ray-tracing the base patterns to $\mathbf{y}_{\text{base}} = P(\text{pose})\, x_{\text{base}}(\mathbf{m}, t)$, the embedded prediction supplied to (32)–(46) is

$$\hat{\mathbf{v}}(\mathbf{m}, t) \;=\; \big( \cos(2\mathbf{y}_{\text{base}}) + \boldsymbol{\delta}^{\cos}(\mathbf{m}, t), \; \sin(2\mathbf{y}_{\text{base}}) + \boldsymbol{\delta}^{\sin}(\mathbf{m}, t) \big). \tag{48}$$

The displayed LCD patterns are read off as $x_{\text{base}}$, while $\hat{\mathbf{v}}$ enters the BFN training objective (46) and the end-back update (47).

**Why a residual around the unit circle.** The trigonometric anchor $(\cos 2\mathbf{y}_{\text{base}}, \sin 2\mathbf{y}_{\text{base}})$ lies on the unit circle $\mathbb{S}^1$, whereas both the BFN belief state $\mathbf{m}$ and the Rao–Blackwellized target $\bar{\mathbf{v}}_{\text{CLF}}$ take values in $\mathbb{R}^2$. In particular, $\bar{\mathbf{v}}_{\text{CLF}}$ has magnitude strictly below one whenever the sign posterior $p(s \mid \mathbf{m}, b)$ in (45) is non-degenerate, and $\mathbf{m}$ is Gaussian with full support in $\mathbb{R}^2$. The additive modulators $(\boldsymbol{\delta}^{\cos}, \boldsymbol{\delta}^{\sin})$ enlarge the predictive support of $\hat{\mathbf{v}}$ from $\mathbb{S}^1$ to $\mathbb{R}^2$, matching the geometry of both the belief distribution and the RB target. The base patterns $x_{\text{base}}$ remain the primary carrier of physical feasibility (softly biased toward the EyeReal valid phase range (13) by the $\sigma(\cdot) - 0.2$ activation) and anchor the embedded prediction at the unit-circle representative $\mathbf{v}(\mathbf{y}_{\text{base}})$; the residual modulators then provide the additional degrees of freedom on the embedded components needed to match the RB target. Through the decoder $\hat{b}(\hat{\mathbf{v}}) = (1 - \hat{v}^{\cos})/2$ the reconstruction term in (46) jointly depends on $x_{\text{base}}$ and $\boldsymbol{\delta}^{\cos}$, allowing the modulator to contribute small intensity corrections beyond what the base patterns alone express, while $\boldsymbol{\delta}^{\sin}$ acts only through the embedded RB term. This residual parameterization is the minimal modification that retains the EyeReal U-Net's phase-domain inductive bias while affording the freedom required by the embedded BFN objective.

**Defaults.** By default we use $K = 3$ LCD layers at the EyeReal resolution, the BFN schedule from (6)–(7) with $\sigma_1 = \sqrt{10^{-3}}$, and the hybrid objective (46) with $\lambda = 10^{-2}$.

## A.6. CLF simulation and physical model

We briefly summarize the forward model used in the CLF task. Let $x$ denote the unknown phase patterns of the three LCD layers (stacked into a single vector), and let pose parameterize the camera/view configuration. The linear operator $P(\text{pose})$ maps layer phases to per-ray phase accumulation $\mathbf{y} = P(\text{pose})x$. The rendered intensity-like observation is then obtained componentwise by the $\pi$-periodic nonlinearity

$$\hat{b}(x; \text{pose}) = \sin^2\left(P(\text{pose})x\right), \tag{49}$$

which matches (12). Because ground-truth $x$ is not available, training is performed by defining targets in the $\mathbf{y}$ domain (Section 4.5.2) so that BFN denoising directly improves the rendered signal $\hat{b}$ toward the observed $b$. At evaluation time, we report reconstruction quality of rendered views (e.g., crosstalk reduction as in Figure 3) together with runtime/real-time constraints.

# B. Multi-seed Results

To assess robustness across random seeds, we rerun the CSP main comparison, the CLF main comparison, and the MP-20 ablations over multiple seeds for all methods and report mean±std. PeriodicBFN remains state of the art on CSP and achieves a strong quality–speed trade-off on CLF (second-best PSNR, Time, and FPS), while the ablations confirm that both the 2D unit-circle embedding and the Rao–Blackwellized objective contribute substantively to performance.

*Table 4.* Multi-seed CSP results on MP-20 and MPTS-52 (mean±std). Best per column in **bold**.

| | MP-20 | | MPTS-52 | |
|---|---|---|---|---|
| | Match rate↑ | RMSE↓ | Match rate↑ | RMSE↓ |
| CDVAE | $33.93 \pm 0.15$ | $0.1069 \pm 0.0018$ | $5.21 \pm 0.13$ | $0.2083 \pm 0.0023$ |
| DiffCSP | $51.89 \pm 0.30$ | $0.0611 \pm 0.0015$ | $13.95 \pm 0.11$ | $0.1422 \pm 0.0053$ |
| EquiCSP | $57.48 \pm 0.18$ | $0.0510 \pm 0.0010$ | $14.48 \pm 0.19$ | $0.1201 \pm 0.0029$ |
| FlowMM | $61.26 \pm 0.14$ | $0.0572 \pm 0.0014$ | $16.11 \pm 0.17$ | $0.1831 \pm 0.0021$ |
| CrysBFN | $64.33 \pm 0.24$ | $0.0445 \pm 0.0010$ | $19.71 \pm 0.72$ | $0.1112 \pm 0.0077$ |
| PeriodicBFN (Ours) | $\mathbf{70.05 \pm 0.04}$ | $\mathbf{0.0370 \pm 0.0009}$ | $\mathbf{27.03 \pm 0.13}$ | $\mathbf{0.1105 \pm 0.0024}$ |

*Table 5.* Multi-seed CLF phase synthesis results (mean±std). Underline = second-best per column.

| Method | PSNR↑ | Time (ms)↓ | FPS↑ |
|---|---|---|---|
| Conjugate Gradient | $34.31 \pm 1.21$ | $691.2 \pm 6.5$ | $1.447 \pm 0.014$ |
| EyeReal | $27.46 \pm 1.18$ | $19.4 \pm 0.25$ | $51.55 \pm 0.53$ |
| EyeReal + PeriodicBFN | $\underline{31.40 \pm 1.05}$ | $\underline{35.1 \pm 0.22}$ | $\underline{28.49 \pm 0.16}$ |

*Table 6.* Multi-seed MP-20 ablations (mean±std).

| Method | Match rate (%)↑ | RMSE↓ |
|---|---|---|
| PeriodicBFN | $70.05 \pm 0.04$ | $0.0370 \pm 0.0009$ |
| w/o RB | $66.11 \pm 0.12$ | $0.0507 \pm 0.0011$ |
| w/o 2D embedding | $60.02 \pm 0.10$ | $0.0569 \pm 0.0015$ |
| w/o $t$ input | $66.17 \pm 0.19$ | $0.0479 \pm 0.0011$ |

# C. Implementation Details

We summarize the architectural choices and training hyperparameters used in our experiments.

**CSP backbone (MP-20 / MPTS-52).** We adopt the CSPNet equivariant backbone with 6 layers, hidden size 512, and 128 sinusoidal frequencies for time/angle embedding, following CrysBFN (Wu et al., 2025). Internally, the noisy 2D unit-circle state $\mathbf{u}_t$ is converted to an angular representation; the network predicts an angle residual and we map back to the 2D embedded space via the unit-circle parameterization. The backbone consumes angular (sin/cos) features, while the

Bayesian belief still retains the 2D radial magnitude as a principled uncertainty carrier. We do not feed the accumulated accuracy as an input (it is implicit through $t$ given strictly additive accuracy).

**CSP training.** We use $\sigma_1$ (the terminal noise-scale parameter) following the BFN convention, 500 sampling steps, the AdamW optimizer, and a cosine learning-rate schedule decaying to a small minimum. We train for 5000 epochs on MP-20 and 3000 epochs on MPTS-52.

**CLF backbone and training.** We use the EyeReal U-Net backbone (Ma et al., 2025) with the embedded output head and hybrid objective described in Section A.5. We train with AdamW and a cosine learning-rate schedule for 6000 epochs on 10,000 stereo pairs rendered from the LEGO bulldozer NeRF scene with random viewer poses and binocular distances $R \in [30, 50]$.

**Code.** The source code, configuration files, and pretrained checkpoints are available at https://github.com/EmperorJia/PeriodicBFN.

## D. Runtime and Compute Overhead

We quantify the practical overhead of the 2D unit-circle representation relative to a 1D circular BFN baseline (CrysBFN) on MP-20 (Table 7). The 2D embedding adds two input channels per periodic scalar but the parameter count is unchanged because the backbone output collapses back to an angular representation. End-to-end training and inference are comparable to (slightly faster than) the 1D baseline; we attribute this in part to the closed-form Rao–Blackwellized target, which avoids extra helper steps required when strict additive accuracy is not available.

*Table 7.* Compute overhead of the 2D embedding compared with a 1D circular BFN baseline on MP-20.

| Metric | PeriodicBFN | CrysBFN (1D circular) |
| --- | --- | --- |
| #Parameters | 12.3M | 12.3M |
| Train time (sec/step) | 0.047 | 0.053 |
| Single-GPU inference, 200 steps (s, after warmup) | 74 | 89 |

## E. Step-count Robustness (NFE Sweep)

We sweep the Number of Function Evaluations (NFE), i.e., the number of network forward passes during sampling. PeriodicBFN saturates at 200–500 forwards, while CrysBFN degrades when the discrete schedule is shortened.

*Table 8.* MP-20 match rate (%) with different NFEs. Both models use 12.3M parameters.

| # Network forwards | CrysBFN | PeriodicBFN |
| --- | --- | --- |
| 10 | 60.18 | 59.13 |
| 200 | 64.35 | 69.78 |
| 300 | 62.04 | 70.05 |
| 500 | 62.14 | 70.06 |

## F. Role of the $t_{\min}$ Threshold

At inference start, the belief encodes zero information, so the angular signal is uninformative and the network output should be effectively angle-free (i.e., the embedded origin). We sample $t \sim \mathcal{U}(t_{\min}, 1)$ for training stability. The probability of drawing $t < t_{\min}$ is $t_{\min}$ itself and the corner case is virtually never encountered during training; it primarily handles the zero-information boundary at the very start of inference. In addition, even without this floor, the first Bayesian update step is scaled by a vanishingly small $\alpha_t$ (see Algorithm 1 and the discrete update rule), so the belief state changes only infinitesimally near $t=0$. Empirically, removing the floor does not noticeably affect generation quality.

