# OpenReview forum: "Periodic Bayesian Flow Networks with Additive Accuracy"
_ICML.cc/2026/Conference — ICML 2026 regular_

### Official Review · Reviewer_5QYP · 2026-02-27

**Soundness:** 3
**Presentation:** 3
**Significance:** 2
**Originality:** 2
**Overall Recommendation:** 5
**Confidence:** 2

**Summary:**

This work addresses the failure of additive-accuracy updates on circular manifolds by embedding each periodic scalar onto a 2D unit circle and performing Gaussian Bayesian updates in the Cartesian space, restoring strictly additive accuracy. It further derives Rao–Blackwellized objectives to analytically marginalize periodic symmetries (global shifts in crystals and π-period ambiguities in phase synthesis), reducing gradient variance.

Key Contributions

Additive-accuracy restoration via 2D unit-circle embedding with Gaussian Bayes updates.

Rao–Blackwellized invariant training objectives for periodic translation and sign symmetries.

Application to crystal structure prediction and compressive light-field phase synthesis, showing improved stability and performance over prior periodic generative models.

Essentially this work provides a stable and modular approach for generative modeling on periodic manifolds.

**Compliance With Llm Reviewing Policy:**

Affirmed.

**Final Justification:**

The rebuttal has addressed my concerns and I raised my score accordingly.

**Key Questions For Authors:**

none

**Limitations:**

Yes

**Strengths And Weaknesses:**

Strengths

Principled treatment of periodicity: Restores additive accuracy via 2D unit-circle embedding, providing a clean probabilistic solution.

Theoretically grounded: Derives Rao–Blackwellized objectives to handle symmetry invariances with reduced gradient variance.

This work demonstrates improvements on crystal structure prediction and phase synthesis tasks. This work also extends Bayesian Flow Networks without major architectural changes.

Weaknesses

This work has domain-specific focus as it is primarily validated on periodic scientific data. It's unclear how general applicable it is  beyond such domains. Another issue is that it provides mostly empirical validation and only limited theoretical guarantees is given beyond additive accuracy restoration.

---

> ### Author Rebuttal · Authors · 2026-03-31
>
> We sincerely thank the reviewer for the supportive evaluation and the constructive feedback.
>
> > **W1:** This work has a domain-specific focus as it is primarily validated on periodic scientific data. It is unclear how broadly applicable it is beyond such domains.
>
> Thank you for raising this important point. We agree that evaluating broader modalities is valuable, and we will add a brief discussion in the revision.
>
> We would like to clarify that our CLF phase synthesis results already provide a strong cross-domain test. CLF is a computational optics and inverse-rendering setting that is substantially different from CSP. Moreover, our formulation uses a unit-circle embedding and Gaussian Bayes updates that are mathematical and **domain-agnostic**. They apply whenever variables live on periodic manifolds with wrap-around boundaries, rather than being tied to a particular scientific domain.
>
> We also agree that it is helpful to spell out additional application areas beyond CSP and CLF. Prior periodic generative modeling has been most visible in scientific settings such as molecular conformations with periodic torsion angles [1]. In the revision, we will add a short discussion of other natural extensions, including robotic joint angles, biological signals with circular features, and periodic time-series forecasting.
>
> [1] Jing, Bowen, et al. "Torsional Diffusion for Molecular Conformer Generation." NeurIPS 2022.
>
> > **W2:** The paper provides mostly empirical validation, and only limited theoretical guarantees beyond additive-accuracy restoration.
>
> Thank you for encouraging clearer theoretical grounding. We would like to emphasize that our framework provides two strict guarantees beyond empirical observations, and we will make this more explicit in the revision:
>
> 1. **Exact forward marginals from additive accuracy.** In continuous-time BFNs, additive accuracy is a structural requirement for a well-defined forward process on periodic variables. Our 2D Cartesian embedding restores additive accuracy and yields an exact closed-form forward marginal, see Eq. 18. This enables continuous-time training without relying on schedule heuristics.
>
> 2. **Guaranteed variance reduction via Rao–Blackwellization.** We appreciate the reviewer noting that our work is “theoretically grounded” via the Rao–Blackwellized objectives. We would like to clarify that this is precisely our formal guarantee beyond additive-accuracy restoration. Our symmetry-invariant targets are conditional expectations obtained by analytically marginalizing latent symmetries, and by the classical Rao–Blackwell theorem their variance is no larger than that of sampling-based symmetry targets. We will make this guarantee explicit in the revision by stating it clearly in Section 4.5 alongside Eq. 26 and Eq. 34.

---

> > ### Author Rebuttal · Reviewer_5QYP · 2026-04-02
> >
> > My concerns have been adequately addressed. Thank you.

---

> > > ### Author Response · Authors · 2026-04-04
> > >
> > > Thank you for reconsidering our paper after the rebuttal. We’re glad our responses addressed your concerns, and we appreciate your review and recommendation.

---

### Official Review · Reviewer_QeYf · 2026-03-08

**Soundness:** 3
**Presentation:** 3
**Significance:** 3
**Originality:** 3
**Overall Recommendation:** 5
**Confidence:** 2

**Summary:**

The paper addresses the challenge of modeling periodic data (data that repeats or "wraps around) such as atomic positions in a crystal or phase patterns in 3D displays.

Core Problem: Traditional Bayesian Flow Networks (BFNs) are highly effective in flat (Euclidean) space because their "accuracy" (the amount of information gained at each step) is strictly additive. However, when applied to periodic data, existing BFN models often use circular distributions (like the von Mises distribution) where accuracy does not add up neatly. This makes the models unstable and reliant on manual tuning to function correctly.

Contributions:

- 2D Unit-Circle Embedding: Instead of modeling periodic data as a 1D scalar that wraps around, PeriodicBFN embeds each value as a point on a 2D unit circle. By performing Bayesian updates in this 2D space, the authors restore strictly additive accuracy, allowing for more stable and predictable training.
- Translation Invariance: The paper introduces a Rao-Blackwellized objective to handle "global periodic translations". In a crystal, shifting all atoms by the same amount results in the same structure; this mathematical trick allows the model to ignore these redundant shifts, which reduces noise and improves learning efficiency.
- New Application to Optics: This is the first work to apply periodic generative modeling to phase synthesis for light-field (CLF) displays, which are used to create glasses-free 3D imagery.
- Experimental Success: The model demonstrates improved stability and performance across both crystal structure prediction and multi-layer phase synthesis tasks compared to previous periodic-data models.

**Compliance With Llm Reviewing Policy:**

Affirmed.

**Final Justification:**

I maintain my Accept rating
I thank the authors for the detailed rebuttal on my concerned and other reviewers concerns.

**Key Questions For Authors:**

1. How does this model compare to modern Diffusion models designed for circles/spheres? Is it actually more accurate, or just mathematically "neater" to train?
2. Invariance vs. Embedding: Which part of your method contributes more to the performance gain: the 2D circle trick (which stabilizes the math) or the Rao-Blackwellized objective (which handles the global shifts)?
3. Since you represent every 1D periodic coordinate as a 2D point, you double the input data size. Does this significantly slow down training or increase memory usage for large systems, like complex crystals?

**Limitations:**

The authors do not include a dedicated limitations section. To improve the paper, they should address the following:

- Dimensionality Overhead: Mapping every 1D periodic value to a 2D coordinate doubles the input size. The authors should discuss if this leads to increased memory usage or slower training for large-scale data like complex crystals.

- Unit-Circle Constraint: During sampling, the model might predict points slightly inside or outside the unit circle. The authors should clarify how they enforce the circular constraint and if "drifting" inside the circle affects data quality.

- Baselines: The paper focuses on comparing different BFN versions. It would be stronger if it explicitly addressed how this method compares to non-BFN standards, such as Riemannian Diffusion, in terms of sampling speed and accuracy.

- Real-world Physics: For the 3D display (CLF) task, it is unclear if the generated "ideal" phase patterns are fully compatible with the physical limitations of actual hardware.

**Strengths And Weaknesses:**

**Strengths**

*Soundness*: The paper is technically rigorous. By shifting from 1D circular distributions (like von Mises) to a 2D Cartesian embedding, the authors provide a mathematically sound way to maintain "strictly additive accuracy." This isn't just a heuristic; it ensures the Bayesian updates remain Gaussian, which is the theoretical bedrock of BFNs. The inclusion of the Rao-Blackwellized objective is a sophisticated way to handle translation invariance, backed by solid statistical reasoning.

*Originality*: While BFNs and circular embeddings exist, the combination is highly original. The paper identifies a specific failure point in prior periodic BFNs (the loss of additivity) and fixes it with an elegant geometric shift. Applying this to Compressive Light-Field (CLF) displays is a novel use case that moves the conversation beyond standard "toy" periodic problems.

*Significance*: The work has high utility for specialized but critical fields. In crystallography and material science, handling wrap-around boundaries is a persistent headache; this model provides a more stable "off-the-shelf" generative tool. Its success in phase synthesis for 3D displays demonstrates broad applicability in physical sciences.

*Presentation*: The paper is well-structured, clearly defining the "additive accuracy" problem early on. It does an excellent job of positioning itself against "Wrapped BFNs" and "vM-BFNs," making the specific mathematical advantages easy to track for a reader familiar with generative modeling.

---

**Weaknesses**

*Soundness*: While the results on crystals and CLF are impressive, the experimental section is somewhat narrow. The paper would be stronger if it compared PeriodicBFN against a wider array of Flow Matching or Diffusion baselines that use periodic manifolds, rather than focusing primarily on beating other BFN variants.

*Significance*: By embedding every periodic scalar into 2D space, the model effectively doubles the dimensionality of the data the neural network must process (e.g., N coordinates become 2N inputs). For massive crystal structures or high-resolution phase maps, this increased computational overhead might limit the model's practical adoption compared to 1D methods.

*Presentation*: The "Rao-Blackwellized" section, while mathematically dense and correct, is significantly more difficult to parse than the rest of the paper. A more intuitive explanation or a toy-example visualization of how this handles "global shifts" would help make the paper accessible to a broader ML audience.

*Originality*: One could argue that the 2D embedding is a known trick in other areas of machine learning (like positional encodings). While its application here solves a specific BFN problem, the "conceptual leap" is more of a clever engineering fix than a fundamental shift in how we understand generative AI.

---

> ### Author Rebuttal · Authors · 2026-03-31
>
> > **Q1/Q5 (Soundness): broader diffusion/flow-matching comparisons on periodic manifolds**
>
> We thank the reviewer for the suggestion. We would like to clarify that our main tables already include strong non-BFN baselines that address the same periodic CSP setting, including diffusion-based methods (e.g., DiffCSP and EquiCSP) and a flow-matching representative (FlowMM). We will make this comparison more explicit and add a short discussion of how these families handle periodicity.
>
> At a high level, diffusion-style periodic methods often model noise with wrapped distributions and require estimating scores on the periodic domain, while flow matching provides a principled alternative but can be sensitive to practical hyperparameter choices. Our approach is based on BFNs; as discussed in [1], BFNs can be viewed through an SDE lens, and one key practical advantage is that the network operates on belief parameters rather than raw noisy samples, which can reduce the effective complexity of the denoising mapping. Empirically, this is reflected in improved accuracy in our updated multi-seed results. Please see the response for W2 & Q1 for Reviewer kePy.
>
> > **Q2/Q7 (Significance): does 2D embedding double compute/memory?**
>
> Thank you for raising this concern. The 2D unit-circle representation adds two input channels per periodic scalar, but the network output can remain a 1D angle/offset that is mapped back to $(\cos\theta,\sin\theta)$. In practice, the overhead is small, and our end-to-end runtime is comparable to (and slightly faster than) a 1D circular BFN baseline.
>
> | Metric | Ours | 1D Circular BFN (CrysBFN) |
> |---|---:|---:|
> | #Parameters | 12.3M | 12.3M |
> | Train time (sec/step) | 0.047 | 0.053 |
> | Single-GPU inference (steps=200, after warmup) | 74s | 89s |
>
> We attribute the efficiency partly to our closed-form Rao–Blackwellized von Mises target, which can be computed directly in the loss, avoiding additional helper steps/tensor operations needed when strict additive accuracy is not available.
>
> > **Q3 (Presentation): make Rao–Blackwellization more intuitive**
>
> We agree. In the revision, we have redrawn Figure 2 (see https://anonymous.4open.science/r/PeriodicBFN-854C) and will further add a simple toy visualization and a more intuitive explanation: the invariant Rao–Blackwellized (RB) target analytically marginalizes over equivalent globally-shifted configurations induced by the periodic symmetry, yielding a lower-variance expected target. We will also discuss its relation to equivariant diffusion (e.g., the visualization in EquiCSP [2]) while emphasizing that our RB objective is derived in closed form.
>
> > **Q4 (Originality): is the 2D embedding just a known trick?**
>
> We thank the reviewer for the thoughtful comment. We agree that unit-circle embeddings are common as *static* input features (e.g., positional encodings). Here, however, the 2D formulation is not a feature-engineering convenience: it resolves a structural issue in BFNs.
>
> Specifically, as formalized in Proposition 4.1, performing Bayesian updates directly on the circle breaks the strictly additive accuracy that BFNs rely on. By moving the belief updates to Gaussian updates in Cartesian $\mathbb{R}^2$, we restore strict additive accuracy and obtain exact continuous-time forward marginals (enabling jumpable training without fragile discrete schedule heuristics). This same formulation also makes our closed-form Rao–Blackwellized objective possible, allowing analytic marginalization of global periodic shifts to reduce gradient variance.
>
> > **Q6 (Invariance vs. embedding): which matters more?**
>
> Both are important and complementary, as shown by our ablation on MP-20.
>
> **Table 4. Ablation studies on MP-20.**
>
> | Method | Match rate (%)↑ | RMSE↓ |
> | :--- | :--- | :--- |
> | PeriodicBFN | 70.05±0.04 | 0.0370±0.0009 |
> | w/o Rao–Blackwellization | 66.11±0.12 | 0.0507±0.0011 |
> | w/o 2D embedding | 60.02±0.10 | 0.0569±0.0015 |
>
> > **References**
>
> [1] Xue, Kaiwen, et al. "Unifying Bayesian flow networks and diffusion models through stochastic differential equations." Proceedings of the 41st International Conference on Machine Learning. 2024.
>
> [2] Lin, et al. "Equivariant diffusion for crystal structure prediction." Proceedings of the 41st International Conference on Machine Learning. 2024.

---

> > ### Author Rebuttal · Reviewer_QeYf · 2026-04-01
> >
> > Done. Thank you.

---

> > > ### Author Response · Authors · 2026-04-02
> > >
> > > Thank you for the constructive feedback and the accept recommendation. We’re glad that our rebuttal resolved your concerns.

---

### Official Review · Reviewer_kePy · 2026-03-10

**Soundness:** 3
**Presentation:** 3
**Significance:** 2
**Originality:** 2
**Overall Recommendation:** 5
**Confidence:** 3

**Summary:**

The paper extends Bayesian Flow Networks to restore additive-accuracy behavior for circular quantities. The method achieves this by embedding each periodic scalar onto the unit circle in 2D and then running standard Gaussian BFN Bayesian updates in that Cartesian space. They show results on crystal structure prediction (MP-20, MPTS-52) and CLF phase synthesis, reporting gains over recent baselines such as CrysBFN and EyeReal.

**Compliance With Llm Reviewing Policy:**

Affirmed.

**Key Questions For Authors:**

### Major
1. Can you report mean±std over multiple seeds for the main tables?

2. Table 1, if bold marks the best value per column then crysBFN MPTS-52 RMSE should be marked in bold.

### Minor
3. The manuscript contains unfinished placeholder “xx%”.

4. Fig 1. has rendering issues (visible compiled lines).

**Limitations:**

No, the authors include an impact statement but no explicit limitations of their framework are discussed

**Strengths And Weaknesses:**

## Strengths
- The contribution is simple and clear
- The results show improvements over recent baselines.

## Weaknesses
- Novelty is somewhat incremental, as the core tricks aren't new, but the authors show and fix a BFN pathology, which is okay.
- Results are reported as single points. It would be more compelling to see mean±std over seeds, to judge robustness and significance.
- Some presentation issues hurt quality (unfinished “xx%” placeholder text, inconsistent bolding in tables, and some figure rendering issues).

---

> ### Author Rebuttal · Authors · 2026-03-31
>
> We sincerely thank the reviewer for the constructive feedback. Below we address the key points and summarize the corresponding revisions.
>
> > **W1: “Novelty is somewhat incremental ...”**
>
> We sincerely thank the reviewer for the thoughtful assessment and for explicitly recognizing that our work “shows and fixes a BFN pathology.” We appreciate the concern that the underlying primitives are not new, and would like to respectfully clarify what we see as the main contributions:
>
> 1. **Core theory.** Our contribution is to restore BFNs’ key property of **strictly additive accuracy** for periodic variables via a principled unit-circle (2D) formulation with Gaussian Bayesian updates. This is not merely an engineering trick, but a theoretical necessity that **unlocks continuous-time training** and **removes reliance on fragile schedule heuristics**.
>
> 2. **Translation invariance.** We derive a **closed-form Rao–Blackwellized objective** that **marginalizes global periodic translations**. In the revision, we mathematically demonstrate that prior circular adaptations (e.g., CrysBFN[1] ) do not readily admit such a tractable, closed-form marginalization due to the non-additivity of their updates.The details are included in the anonymous supplementary appendix https://anonymous.4open.science/r/PeriodicBFN-854C.
>
> 3. **Empirical scope.** We not only validate our approach on standard CSP benchmarks but, more importantly, extend periodic generative modeling to **CLF phase synthesis**. This demonstrates that our theoretical fix scales effectively to complex, high-dimensional inverse rendering problems in modern optics.
>
> [1] Wu, Hanlin, et al. "A Periodic Bayesian Flow for Material Generation." The Thirteenth International Conference on Learning Representations.
>
> -----
>
> > **W2 & Q1: Report mean±std over multiple seeds.**
>
> We agree. We reran **all** methods in the main tables (all baselines included) over multiple seeds for both CSP and CLF, and now report mean±std. PeriodicBFN is **SOTA** on MP-20 and MPTS-52, and achieves a strong **quality–speed** trade-off on CLF (second-best across PSNR/Time/FPS).
>
> **Table 1 (updated): MP-20 / MPTS-52 (CSP)**
>
> |Method|MP-20 Match↑|MP-20 RMSE↓|MPTS-52 Match↑|MPTS-52 RMSE↓|
> |---|---|---|---|---|
> |CDVAE|33.93±0.15|0.1069±0.0018|5.21±0.13|0.2083±0.0023|
> |DiffCSP|51.89±0.30|0.0611±0.0015|13.95±0.11|0.1422±0.0053|
> |EquiCSP|57.48±0.18|0.0510±0.0010|14.48±0.19|0.1201±0.0029|
> |FlowMM|61.26±0.14|0.0572±0.0014|16.11±0.17|0.1831±0.0021|
> |CrysBFN|64.33±0.24|0.0445±0.0010|19.71±0.72|0.1112±0.0077|
> |**PeriodicBFN (Ours)**|**70.05±0.04**|**0.0370±0.0009**|**27.03±0.13**|**0.1105±0.0024**|
>
> **Table 3 (updated): CLF phase synthesis**
>
> Underlined = 2nd-best per column.
>
> |Method|PSNR↑|Time(ms)↓|FPS↑|
> |---|---|---|---|
> |Conjugate Gradient|**34.31±1.21**|691.2±6.5|1.447±0.014|
> |EyeReal|27.46±1.18|**19.4±0.25**|**51.55±0.53**|
> |**EyeReal + PeriodicBFN**|$\underline{31.40\pm1.05}$|$\underline{35.1\pm0.22}$|$\underline{28.49\pm0.16}$|
>
> -----
>
> > **W3 & Q4: Figure rendering issues.**
>
> We apologize for the artifacts. We have redrawn Figures 1–2 and replaced them in the revised paper; updated figures are also available at https://anonymous.4open.science/r/PeriodicBFN-854C. Figure 1 illustrates the two application settings and their inherent periodic symmetries (CSP fractional coordinates; CLF phase ambiguity). Figure 2 summarizes the PeriodicBFN pipeline, highlighting the unit-circle (2D) representation and the belief-update procedure.
>
> -----
>
> > **Q2: Table 1 bolding (MPTS-52 RMSE).**
>
> Thank you for pointing this out. In the updated multi-seed results, PeriodicBFN has a slightly lower mean MPTS-52 RMSE than CrysBFN (0.1105±0.0024 vs. 0.1112±0.0077), so the bolding is correct in the revision.
>
> -----
>
> > **Q3: Unfinished placeholder “xx%”.**
>
> We apologize for the oversight. We removed the placeholder and now report the exact gains (e.g., +5.7 and +7.3 match-rate points over the strongest baseline in Table 1).
>
> -----
>
>
>
> > **Limitations.**
>
> We agree and will add an explicit limitations paragraph in the revised version. For CLF, our iterative updates (though already real-time) can be slower than one-shot predictors like EyeReal; future work will explore lighter periodic-aware backbones to further improve efficiency.

---

> > ### Author Rebuttal · Reviewer_kePy · 2026-04-01
> >
> > Thank you, my concerns have been addressed.

---

> > > ### Author Response · Authors · 2026-04-02
> > >
> > > Thank you for your review and the the accept recommendation. We are pleased that our rebuttal has addressed your concerns.

---

### Official Review · Reviewer_nR3o · 2026-03-13

**Soundness:** 3
**Presentation:** 3
**Significance:** 3
**Originality:** 3
**Overall Recommendation:** 5
**Confidence:** 3

**Summary:**

This work focuses on an improvement to the use of bayesian flows for modeling and generating data for periodic systems.  The authors highlight that previous methods for modeling periodic systems with bayesian flows use circular distributions, which are not strictly additive and therefore break an underlying assumption of bayesian inference.  This work, PeriodicBFN maps periodic scalars to a unit circle in R2, a latent space in which the additivity required by bayesian flows is maintained.  The authors derive this embedding, and the necessary objectives and loss functions under a gaussian belief family.  The embedding uses a task-dependent rotational symmetry, for which embedding targets are evaluated mathematically for two tasks:  Crystal structure prediction (CSP) and compressive light field (CLF).  Improvements in both tasks are presented as compared to baseline methods.

**Compliance With Llm Reviewing Policy:**

Affirmed.

**Final Justification:**

My concerns were addressed during rebuttal.  I maintain my Accept rating

**Key Questions For Authors:**

Is the tmin threshold ever active during training or inference?  Do the authors have data for how necessary this correction is practically, and how well the model performs in such cases?


The authors claim “This is not an arbitrary extra degree of freedom: the radial magnitude is a natural carrier of uncertainty under the Gaussian belief parameterized by ρ(t).“  If such a case is true, could the authors apply an ablation under which t is not included as a network input?

**Limitations:**

yes

**Strengths And Weaknesses:**

Strengths:
The authors correctly identify a weakness in the use of existing methods for modeling periodic data using bayesian flows and develop a mathematically rigorous method for solving this problem.

The authors effectively demonstrate the efficacy of their method on two tasks with periodic data.

The writing is of high quality, clear and readily understandable.


Weaknesses:


Figure 1 is generally of poor quality, with unclear visuals, overlapping boxes, partial edges on portions of the figure and unclear connections between the various diagrams.

The authors do not provide code, hyperparameters, or a detailed breakdown of network structure, weakening  reproducibility

Minor errors:
Line 134: “descrete”

In Table 1, MPTS-52, RMSE column, PeriodicBFN is bolded over the better CrysBFN result

Figure 2 includes multiple typos

---

> ### Author Rebuttal · Authors · 2026-03-31
>
> We thank the reviewer for the positive assessment. We address each point below and have incorporated the fixes in the revision.
>
> > **W1: Figure 1**
>
> We apologize for the visual artifacts in the submitted version. We have **redrawn Figures 1–2** and replaced them in the revision; the updated figures are also available at https://anonymous.4open.science/r/PeriodicBFN-854C.
>
> - Figure 1 illustrates the two periodic application settings and their symmetries: CSP (fractional coordinates) and CLF (phase ambiguity).
> - Figure 2 summarizes the PeriodicBFN pipeline, emphasizing the unit-circle (2D) representation, Bayesian belief updates and Periodic training.
>
> > **W2: Reproducibility: code, hyperparameters, and network details**
>
> We agree and have released our implementation at https://anonymous.4open.science/r/PeriodicBFN-854C. Key settings:
>
> **CSP (MP-20 / MPTS-52).**
> - Backbone: CSPNet with 6 layers, 512 hidden size, and 128 frequencies for sinusoidal embedding, following CrysBFN [1].
> - Internal angle conversion: convert noisy $(x_t, y_t)$ to $\theta_t=\mathrm{atan2}(y_t,x_t)$; predict an angle residual; map back via $(\cos\hat\theta_0,\sin\hat\theta_0)$.
> - Backbone uses **angular information** (sin/cos), while the **Bayesian belief** still retains the 2D radial magnitude as a principled uncertainty carrier.
> - Hyperparameters: $\sigma_1^2=0.001$; sampling steps = 500; optimizer AdamW with lr $1\times 10^{-3}$; cosine schedule with min lr $1\times 10^{-5}$; epochs = 5000 (MP-20) and 3000 (MPTS-52).
> - We remove the “accumulation accuracy” input due to strictly additive accuracy.
>
> **CLF.**
> - Backbone: EyeReal U-Net [2].
> - Input: concatenate warped noisy views + warped conditioning views + 128-d sinusoidal time embedding ($D_{in}=164$).
> - Hyperparameters: AdamW lr $1\times 10^{-3}$; cosine LR; 6000 epochs on 10k stereo-pairs (LEGO bulldozer).
>
> > **W3: Minor typo (Line 134: “descrete”)**
>
> Thank you. We have corrected “descrete” to “discrete” in the revision.
>
> > **W4 Table 1 boldface error (MPTS-52, RMSE)**
>
> Thank you for catching this. In the submission, the bolding in the MPTS-52 RMSE column was inconsistent. In the updated multi-seed results (mean±std), PeriodicBFN has a slightly lower mean MPTS-52 RMSE than CrysBFN ($0.1105\pm0.0024$ vs. $0.1112\pm0.0077$), so bolding PeriodicBFN is correct in the revision. Please see W2 & Q1 in the response for Reviewer kePy
>
> > **W4: Figure 2 typos**
>
> Thank you. We have corrected the typos and improved the wording/labels for clarity in the redrawn Figure 2.
>
> > **Q1: Is $t_{\min}$ ever active? Is it practically necessary?**
>
> Great question. Even if we sampled $t\sim\mathcal{U}(0,1)$ without enforcing a lower bound, the probability of drawing $t<10^{-10}$ is $10^{-10}$, so this corner case is virtually never encountered during training; it mainly matters at the **zero-information boundary** at inference start.
>
> From a BFN perspective, the network is conditioned on the current belief (prior) and produces an observation/sample for the next update. At $t=0$, the belief contains zero information, so the network should output a fixed, angle-free vector, i.e., the origin $(0,0)$ (no directional preference).
>
> Importantly, even if the network were not forced to output exactly $(0,0)$ at $t=0$, the practical effect would still be negligible: the first Bayesian update step from $t=0$ is scaled by a very small $\alpha$ (see Algorithm 1 and Eq. (6)), so the belief changes only infinitesimally at the start. We will clarify this intuition and the role of $t_{\min}$ around Eq. (20).
>
> > **Q2 Ablation: removing $t$ from the network input**
>
> Thank you for the insightful suggestion. We ran this ablation on MP-20:
>
> | Setting | Match rate | RMSE |
> |---|---:|---:|
> | PeriodicBFN | $70.05\pm 0.04$ | $0.0370\pm 0.0009$ |
> | w/o $t$ input | $66.17\pm 0.19$ | $0.0479\pm 0.0011$ |
>
> This supports the intuition that radial magnitude carries uncertainty in expectation, but also shows a practical issue: a single sampled $\mu_t$ provides only a **noisy** cue of the true noise level. Without explicit $t$, the backbone must infer the denoising scale from this noisy cue, degrading denoising. Providing $t$ supplies the exact schedule signal and cleanly decouples uncertainty (belief) from control (denoising scale).
>
> [1] Wu, Hanlin, et al. “A Periodic Bayesian Flow for Material Generation.” The Thirteenth International Conference on Learning Representations.
>
> [2] Ma, Weijie, et al. “Glasses-free 3D display with ultrawide viewing range using deep learning.” Nature 648.8092 (2025): 76-83.

---

> > ### Author Rebuttal · Reviewer_nR3o · 2026-04-02
> >
> > Thank you for your detailed response.  My concerns have been addressed

---

> > > ### Author Response · Authors · 2026-04-02
> > >
> > > Thank you for the positive evaluation. We’re glad the rebuttal addressed your concerns.

---

### Decision · Program_Chairs · 2026-04-30

**Decision:**

Accept (regular)

**Comment:**

The paper identifies a limitation of prior periodic BFNs and proposes a simple but principled fix via unit-circle embedding with additive-accuracy restoration, further strengthened by a Rao–Blackwellized invariant objective. Reviewers agreed the method is technically sound, clearly presented, and empirically strong on both crystal generation and CLF phase synthesis. The main concerns (incremental novelty, domain specificity, and modest 2D overhead) were all satisfactorily addressed in the rebuttal and do not appear decision-critical. I therefore recommend accept.